

# The role of forest maturity on catchment hydrologic stability

Oscar Belmar [1], José Barquín [1], Jose Manuel Álvarez-Martínez [1], Francisco J. Peñas [1], Manuel Del Jesus [1]

[1] Environmental Hydraulics Institute, Universidad de Cantabria - Avda. Isabel Torres, 15, Parque Científico y Tecnológico de Cantabria, 39011, Santander, Spain

*Correspondence to*: Oscar Belmar (oscar.belmar@unican.es)

**Abstract.** Land cover and soil properties largely determine how climatic and hydrological regimes interact and produce hydrological stress in aquatic ecosystems. This study aims to clarify the influence of forests, as well as other majoritarian land cover types, on hydrological regime through an experimental design without the main limitations associated with traditional paired-watershed studies. With this aim, we use more catchments and an additional forest descriptor: forest maturity. We focus on flood and drought regimes, as they constitute the extremes of hydrological variability. Specific objectives were to isolate the relative contribution of precipitation and land cover composition to such flow extremes and to contrast the effectiveness of forests (surface and maturity) and other land cover types to predict them. The study was developed in a heterogeneous region located in the Cantabrian Mountains (NW Spain) with different vegetation types and a long history of human disturbance and land use change that allowed a robust experimental design. Regression and partial correlation analyses were developed using hydrological and meteorological data combined through hydrological modelling using IHACRES. Land cover characteristics showed ability to predict both flood regimes and low flows, although low flows were explained mainly by precipitation regimes. Forests showed a stabilization effect on flow regime (lower floods and greater base flows), but the effect was more evident with forest maturity than with surface. Other land cover types showed different effects. Evaluating the role of land cover on hydrological stability requires the use of comprehensive information involving different descriptors and their temporal changes, not only the current surface occupied by each land cover type.

**Keywords.** Cantabrian Mountains, Native Forests, Maturity, Flow regime, Land use, River disturbance, IHACRES

## 1 Introduction

Flood and drought events represent the extreme demonstration of hydrologic variability, which constitutes a primary driver of stream biological communities and ecosystem functioning (Resh et al. 1988; Lake 2000). Such events may cause greater impacts on river ecosystems than changes in average conditions (Woodward et al. 2016), and their frequency, intensity and duration are expected to increase due to Climate Change (IPCC 2012). In addition, land use changes may also affect flood and drought phenology (e.g. Scott and Lesch 1997), and therefore their consequences on ecosystems.

Forest area has increased in Europe over the last decades (Spiecker et al. 2012). Socioeconomic adjustments, such as those linked to the EU Common Agricultural Policy (CAP), have led to a dramatic rural exodus and subsequent abandonment of



agricultural land, a cessation of coppicing and a reduction in grazing in natural communities (e.g.: Benayas et al. 2007). Today, forests cover nearly 40% of the European surface (European Commission 2015). Their trees have greater water requirements than other vegetation types, as they intercept more precipitation and present greater transpiration rates (e.g. Bosch and Hewlett 1982). Thus, their expected effect on river flows would be a general reduction when forests spread, grow

and mature (Johnson 1998).

The development of paired-watershed experimental designs has aimed to clarify forest influence on the water cycle (Hewlett 1971, 1982; Cosandey 1995). These studies are generally based on selecting two similar and geographically close watersheds, subjected to the same climatic regime, and assuming that different hydrologic responses will be driven by differences in forest extent. Their results generally indicate that the effect of forest expansion is a decrease in water yield

(both flood volumes and peaks). Nevertheless, additional watershed-scale research is necessary to advance our understanding of forest impact on hydrology. Particularly, studies focused on large basins (several tens of km2), additional descriptors of forest characteristics (besides area) and a larger number of observed watersheds (instead of only two) (Andréassian 2004). More complete studies that clarify the relationship between forests and hydrological processes may allow improving the design of strategies to face the effects of Climate Change on catchment hydrology. In this context, forest maturity may be an

important factor to determine forest-river flow relationships, as the long process of native forests formation involves many steps that increase water retention. Tree roots grow into fissures and aid in the breakdown of bedrock, penetrating compacted soil layers and allowing soil aeration and water infiltration. In addition, a vegetative ground cover modifies the temperature and moisture conditions below (Fisher and Stone 1969; Fisher and Eastburn 1974). Given the interaction of these processes with the hydrological cycle, the use of maturity as a descriptor of forest characteristics in experimental designs may improve

our understanding of forests' influence on river ecosystems at a watershed scale.

The aim of this study is to improve the understanding of how forests and other predominant land cover patches influence flood and drought patterns at a watershed scale through an experimental field design without the limitations of paired-watershed studies. We use several large catchments in the Cantabrian Mountains (NW Spain) with a gradient of forest cover due to human management since the 15th century. We defined forest cover not only through forest surface but also using

forest maturity. Our specific objectives are: (i) to isolate the relative contribution of precipitation regimes and land cover to hydrological extreme events and (ii) to compare the effectiveness of forest, as well as other predominant land cover types, surface and maturity to predict such extremes. We expect mature forests to smooth hydrological extremes caused by precipitation regimes through water interception and retention, in opposition to other land cover types. Thus, forest maturity is expected to be negatively associated with floods and positively related to base flows, inducing hydrologic stability. Such

outcome would be relevant for environmental management in order to face the effects of Climate Change in river ecosystems at catchment scale.





## 2 Material and methods

### 2.1 Study area

This study has been developed in the Cantabrian Mountains, which extend for more than 300 km across northern Spain, nearly parallel to the Cantabrian Sea. They constitute a distinct physiographic province of the larger Alpine System

physiographic division. Glaciers and fluvial erosion are the two main processes that have shaped their relief, composed mainly of sedimentary materials such as limestone and conglomerates. These mountains present an Atlantic climate with annual precipitation and temperature around 1160 mm and 9.5 °C, respectively. Areas located at lower latitudes show sub-Mediterranean characteristics, with higher temperatures and summer droughts (Ninyerola et al 2007). This environmental heterogeneity shelters a mix of tree species with beeches (Fagus sylvatica), birches (Betula ssp.) and different species of oaks

(*Quercus petraea*, *Q. robur*, *Q. pyrenaica* and *Q. rotundifolia*) in a transition from the Atlantic to the sub-Mediterranean areas. Shrub vegetation spans a similar gradient, varying from semi-arid communities mixed with annual grasslands and crops in the southeast to shrubs and young forests in the north and west, with alpine vegetation and bare rock at higher elevations and slopes.

A set of 16 catchments, later reduced to 10 due to data quality issues (see Meteorological and hydrological data; Fig. 1,

Table 1), was selected to represent a land cover gradient within a similar climatic region. Such gradient characterizes the legacy of human management and land use practices for the last 400 years. After the foundation of the Real Fábrica de Artillería de la Cavada (in English, the Royal Artillery Factory in La Cavada) in 1616, the native forests in the Eastern extreme were intensively exploited for more than 200 years in order to obtain wood for naval construction. Since then, this area has been kept deforested for stockbreeding through the combined use of fire and cattle grazing. Consequently, the

Eastern part of the study area is dominated by a mixture of shrubs with a dominance of dry heathland communities and extensive pastureland. Only some isolated patches of forest remain in steep hillslopes. On the contrary, the western catchments have not suffered massive deforestation and still present quite well developed mature forests. The presence of brown bear (*Ursus arctos*) and Cantabrian Capercaillie (*Tetrao urogallus cantabricus*) in these catchments, unlike the eastern extreme (González et al. 2016; Blanco-Fontao et al. 2012), evidences a better state of conservation.

### 2.2 Land cover characteristics

Land cover information was obtained from a supervised classification approach of remote sensing imagery. A suitable Landsat TM image of the study area taken in 2010, with a minimum cloud cover and a relatively high sun elevation angle, was downloaded from the United States Geological Survey (USGS). The image was radiometrically and atmospherically corrected using the algorithms available in GRASS (2013). A complementary digital elevation model (DEM) was obtained

from LiDAR data (CNIG 2014) and resampled to 30 meters to match the spatial resolution of the image. A per-pixel classification approach was applied using a Maximum Likelihood algorithm (ML) over a combination of spectral information and topographic layers derived from the 30-m DEM. The ML algorithm assigned pixels to the land cover class



with maximum membership probability, although they may have an almost equal probability of membership to another class (Lewis et al. 2000). Simultaneously, a fuzzy k-means classification allowed yielding membership probabilities for each land cover type at the pixel level (see more details in Álvarez-Martínez et al. 2010). The relative surface occupied by each land cover class in each catchment was obtained using the result of the ML algorithm. However, forest maturity was estimated

using the membership probabilities obtained through the fuzzy classification, calculating the average per-pixel forest probability in each of the selected catchments. Pixels with a higher forest probability represent dense forest patches that can be interpreted like developed, mature forests. Higher forest probability can be interpreted like greater forest maturity.

## 2.3 Meteorological and hydrological data

Meteorological records were acquired from the Spain02 database (version 4), developed by the Agencia Estatal de
Meteorología (AEMET, the State Meteorological Agency) and the Universidad de Cantabria (UC, University of Cantabria). Such database includes gridded datasets interpolated with rainfall and temperature data from over 2500 stations in Spain at different resolutions for the period 1971-2007 (Herrera et al. 2012; 2016). Meteorological series (rainfall and temperature) were obtained by averaging those cells belonging to the grid within each catchment. The resulting rainfall and temperature series were represented using box-plots in order to verify that the catchments in the study area presented reasonably similar
climatic regimes.

Flows recorded by the Red Oficial de Estaciones de Aforo (ROEA, the Official Network of Gauging Stations) were obtained from the Anuario de Aforos database available online at the Centre for Studies and Experimentation on Public Works (CEDEX, 2016). Only the gauging stations located at the outlet of each sub-basin were retained. After testing flow records in order to detect deficiencies (see details in Peñas et al. 2014), 6 out of the initial 16 stations (and their corresponding sub-
basins) were discarded from the study: one in the Duero Basin (2034; Fig. 1) and the five in the Ebro Basin (9092, 9178, 9202, 9203 and 9254; Fig. 1). The flow series of the remaining 10 stations were divided by the mean to allow inter-basin comparison (Poff et al 2006).

## 2.4 Hydrological analyses

Three flow indices were chosen, based on those used by Olden and Poff (2003): (i) the maximum 3-day mean annual flow
(M3DMF); (ii) the mean number of high flow events per year using an upper threshold of 9 times the median flow over all years (FRE9) and (iii) the Base Flow Index (BFI, the seven-day minimum flow divided by mean annual daily flow averaged across all years). The latter was used to characterise low-flow conditions, whereas the two others were used to characterise flood regimes (magnitude and frequency). The period selected for computation was 1995-2010, in order to ensure 15 years of records (Kennard et al. 2010) and match the timing of the image taken by the USGS. These indices were also calculated
using contemporary precipitation series (P-M3DMF, P-FRE9 and P-BFI).

Water interception and retention caused by ground vegetation and soil development were estimated determining the proportions of slow and quick flows through a physical model that uses precipitation, temperature and flow data: IHACRES.





A detailed description is provided in Jakeman and Hornberger (1993). It is composed of a non-linear loss module that converts precipitation to effective precipitation and a linear routing model that converts effective precipitation to streamflow. The non-linear module comprises a storage coefficient (c), a time constant for the rate of drying (tw) of the catchment at a fixed temperature (20 ℃) and a factor (f) that modulates for changes in temperature. A configuration of two parallel storages

in the linear routing module was implemented. By doing so, the proportional volume of the quick flow ($\upsilon q$) to slow flow ($\upsilon s$) storage response for each of the 10 selected catchments was obtained through calibration using the period for which our data were best (2000-2007).

**2.5 Analysis of the effect of climate and land use on hydrological regime**

The effects of precipitation and land cover on the selected hydrological indices were isolated through partial correlation

analysis. The three hydrological indices (M3DMF, FRE9 and BFI) were predicted through Ordinary Least Square (OLS) regression modelling using the three precipitation indices (P-M3DMF, P-FRE9 and P-BFI). Then, by means of new OLS regression modelling, we explored whether land cover characteristics predicted the hydrological variance not explained by precipitation indices (i.e. the residuals of the first model run). In order to contrast the effectiveness of different land cover characteristics to predict hydrological extremes and water interception and retention, land cover characteristics were used to

predict the hydrological indices and the proportional volumes of the quick ($\upsilon q$) and slow ($\upsilon s$) flows through a new OLS regression modelling.

Dependent variables were transformed to reduce heteroscedasticity (King et al. 2005), using decimal logarithms for flow indices and the arcsine of the squared root for the volumes of quick and slow flows, as they were proportions (McDonald 2014). All analyses were developed using the R software (version 3.1.3; R Core Team 2015) through the base package

"stats".

**3 Results**

The catchments in the western extreme showed larger areas covered by forests as well as higher values of forest maturity, whereas those in the eastern extreme presented the lowest values of both parameters. Shrub surface showed the opposite pattern. Pastureland did not show clear differences across the study area (Table 1). The ten studied catchments displayed

reasonably similar climatic regimes, with only a very subtle gradient in climatic characteristics from West to East (Fig. 2).

Only the Base Flow Index showed correlation with precipitation regimes (Table 2). On the contrary, the hydrological indices associated with the magnitude and frequency of floods (M3DMF and FRE9) showed a very low correlation with precipitation. Land cover characteristics, particularly shrub surface and forest maturity, showed a significant relationship with M3DMF and FRE9 after removing the effect of precipitation. In all partial correlations, forest maturity showed higher

correlation scores with hydrological indices than forest surface.



Forest maturity and shrubs surface showed the highest ability for hydrological extremes prediction (Fig. 3). Forest maturity showed a negative relationship with the magnitude and frequency of floods and positive with the base flow, while shrub surface showed opposite trends. Forest maturity and shrubs surface also showed the highest ability for quick/slow flow prediction, as the R2 values and p-values of the OLS modelling evidenced (Fig. 4). Slow flows were positively correlated

with forests maturity and negatively with shrubs. Therefore, like in the partial correlations, forest maturity performed better than forest surface in hydrological prediction.

## 4 Discussion

This study shows that land cover may be more relevant than precipitation to determine the spatial variability of flow extremes in similar, close catchments and points out forest maturity as one key factor to explain hydrological variability,

more effective than forest surface. We reckon that the complex land cover mosaic and land cover dynamics of the selected region in the Cantabrian Mountains (NW Spain) has largely contributed to produce statistically significant modelling results even with a relatively low number of cases. We consider that such results have implications for water management in areas with similar climate, land cover types and land uses (i.e. in temperate Atlantic catchments).

### 4.1 Precipitation and land cover contribution to flow extremes

Precipitation and land cover have different relative contributions to floods and droughts, as partial correlations showed. The spatial variation of floods is determined mainly by land cover characteristics, which implies a huge effect of the land cover mosaic on torrentiality at a catchment scale. Droughts are explained mainly by precipitation regimes. This means that land cover characteristics are unable to overcome the impact of critical reductions in water incomes. However, a moderate influence of land cover on droughts has also been observed, as land cover characteristics showed ability to predict also low

flows (not only floods).

This study indicates that mature forests confer hydrological stability to river flows. Catchments with higher forest maturity presented less intense and frequent floods and greater base flows. Forest maturity also predicted the spatial variability of slow and quick flows better in the selected catchments. Croke et al. (2004) observed the same pattern between forests and the proportional volume of quick and slow flow storage using surface. However, they obtained their results in a relatively small

sub-basin through simulation by combining a generic crop model (CATCHCROP; Perez et al. 2002) with IHACRES. The use of several (and larger) catchments (as Andréassian 2004 suggested) and estimates both of forest surface and maturity in this study based on real data provides more reliable results. Given the higher performance obtained, the use of forest maturity determined by forest probability obtained through fuzzy-logic approaches (see Álvarez-Martínez et al. 2010) may provide an income for hydrologic modelling capable of producing better results. This is especially relevant for water

research due to the widely spread use of vegetation surface in such modelling (e.g. Soil and Water Assessment Tool, SWAT; Arnold et al. 1998).



### 4.2 Implications for forest management

The role that mature forests may play conferring hydrological stability at a catchment scale is unlikely to be accomplished by reforestations. Frequently, reforestation efforts are based on a comparatively small number of fast-growing exotic species. These species have particular environmental preferences and, not surprisingly, many do not always grow as well as expected (e.g. Lamb et al. 2005). Reforestations are then likely to lack a developed ground vegetation cover and a mature soil (at least, during the first decades). They will be less effective in intercepting and retaining precipitation, and therefore, in providing base flows. Instead, their water consumption may contribute to water scarcity and aridification (Jackson et al. 2005; Brown et al. 2005; Sun et al. 2006). Consequently, whereas native forest expansion and maturation could constitute a powerful strategy to counteract the effects of Climate Change in the mid- to long-term through flow regime stabilization, the deterioration of native forests due to human land use management in a context of Climate Change will exacerbate its effects and constitute a drawback impossible to undo in the short term.

### 4.3 The importance of recent past in land cover mosaic

The different performance in hydrologic prediction showed by forest surface and maturity may be related to landscape dynamics. Besides the exploitation that the forests in the study area experienced since the 15th century, the Cantabrian Mountains have showed a major decline in livestock grazing pressure for the last 40 years (Morán Ordóñez et al. 2011, Álvarez-Martínez et al. 2013). This has resulted in a displacement of shrubs and pastureland by native forests in many different areas (Poyatos et al. 2003; Álvarez-Martínez et al. 2014). In our case, more than 10% of pixels classified as forest in 2010 were pasture or shrub in 1984 (unpublished data). This involves that new forest surface comprises a mixture of pixels with different degree of development, and with different hydrological effect at a catchment scale. New forest pixels would present then reduced ground vegetation, organic matter decomposition and soil development (Binkley and Fisher 2012) than if they were occupied by mature forests. We believe that this might be the main reason why forest surface has a poorer ability to explain the spatial variability of hydrological indices, while forest maturity performed much better.

Similarly, the different performance showed by other land cover types not associated with forests also indicates an influence of land cover dynamics on hydrological response. Pastureland was not a good predictor, whereas shrub surface was highly related to hydrological instability. The lack of relationship between pastureland and hydrological indices could be caused by the low representation of pastures in the study area in comparison with dominant land cover types (i.e. forests and shrubs). On the contrary, the better performance of shrubs may be related to land use management, which makes shrub lands a dominant land cover through the extensive and recurrent use of fire (Pausas and Fernández-Muñoz 2012; Regos et al. 2015). Commonly, the shrubs in the study area present a pattern of degraded vegetation and poor soil structure associated with recurrently burnt areas (Díaz-Delgado et al. 2002; Gimeno-García et al. 2007). In this context, the development of additional land cover descriptors, such as maturity for forests, remains necessary to explore the effects of land cover mosaics on hydrological response at a catchment scale.



Finally, further research on the long-term impacts of land cover on hydrologic regimes at a catchment scale may provide key guidelines for a sustainable land use management. First, through the development of more complex designs based on land cover dynamics. For example, using data from different years during the last decades. By doing so, the changes in climate, flows and land cover could be quantified and compared in order to determine the relative contribution of landscape dynamics to hydrological change during the selected period (with a complete understanding of the performance of land cover characteristics through the observation of their changes during those years and some on ground measurements when possible). Unfortunately, such analyses were not possible in our study area due to data availability and quality issues. Second, with hydrologic models based on additional land cover descriptors to extension (such as maturity, for forests). That would allow mimicking land cover-hydrology interactions more accurately. Finally, we consider that understanding the physical mechanisms that may explain the interactions herein observed is mandatory. The influence of tree physiological conditions (e.g. basal area, live biomass or leaf area) deserves special attention, considering the impressive water holding capacity of O horizons (for example, a 5 cm thick O horizon in a sub-alpine forest might have a mass of about 5 kg m -2 and could retain about 10 litres of water; Golding and Stanton 1972). By doing so, we will be able to better assess the contribution of native forests and native forest soils to flow regimes through base flows at a catchment scale, as well as the contribution of other land cover types to quick flows.

## 5 Author contributions

OB performed research, analysed data and wrote the paper. JB conceived the study, performed research and contributed to analyses and writing. JMAM performed research and contributed to analyses and writing. FJP contributed to analyses and writing. MDJ performed research and contributed to writing.

## 6 Competing interests

The authors declare that they have no conflict of interest.

## 7 Acknowledgements and Funding Information

We wish to thank the Ministry for Economy and Competitiveness for its support to the projects "Land use legacy effects on river processes: implications for integrated catchment management (RIVERLANDS; BIA2012-33572)" and "Effects of Hydrological Alteration on River Functioning and Service Provisioning: Implications for Integrated Catchment Management (HYDRA; BIA2015-71197)". We also thank AEMET and UC for the data provided (Spain02 v4 EURO-CORDEX dataset, http://www.meteo.unican.es/datasets/spain02) and Joaquin Bedia for his advice on climatic data processing. Mario Álvarez-Cabria, Ana Silió, Edurne Estévez, Alexia González-Ferreras, Tamara Rodríguez-Castillo, María Lezcano, Ignacio Pérez-



Silos and Mireya Cayon revised an early draft of the manuscript. Finally, we also wish to thank the Ministry for Economy and Competitiveness for its financial support to José Barquín and Oscar Belmar through the Ramón y Cajal (RYC-2011-08313) and Juan de la Cierva (FPDI-2013-16141) programs, respectively.

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





**Table 1. Characteristics of the selected catchments in the Cantabrian Mountains ordered from West (top) to East (bottom). Gauge codes and main river names are provided in the first column.**

| Code (Name) | Geologic and hydrologic characteristics | | | | | Land cover characteristics (%) | | | |
|---|---|---|---|---|---|---|---|---|---|
| | Area (km$^2$) | Altitude (m) | Slope (%) | Mean runoff (hm$^3$) | Mean flow (m$^3$/s) | Forest maturity | Forest surface | Shrub surface | Pasture surface |
| 1296 (Ponga) | 34 | 1277 | 29 | 55 | 2 | 82 | 62 | 33 | 3 |
| 1295 (Sella) | 480 | 1005 | 29 | 627 | 18 | 75 | 41 | 42 | 6 |
| 1274 (Cares) | 266 | 1454 | 31 | 245 | 8 | 72 | 15 | 32 | 9 |
| 2035 (Besandino) | 70 | 1498 | 19 | 33 | 1 | 52 | 7 | 45 | 17 |
| 1265 (Deva - O.) | 296 | 1185 | 26 | 141 | 4 | 77 | 39 | 35 | 12 |
| 1268 (Deva - P.) | 648 | 1029 | 27 | 415 | 15 | 75 | 38 | 38 | 11 |
| 1264 (Bullón) | 156 | 972 | 25 | 63 | 2 | 78 | 56 | 31 | 10 |
| 1215 (Pas) | 358 | 599 | 19 | 270 | 9 | 55 | 33 | 57 | 7 |
| 1207 (Miera) | 161 | 563 | 21 | 147 | 5 | 48 | 22 | 64 | 9 |
| 1196 (Asón) | 492 | 558 | 20 | 649 | 22 | 62 | 32 | 47 | 14 |

**Table 2. Squared R-values obtained for (left) regression modelling of hydrological indices (M3DMF: maximum flow, FRE9: high flow events, BFI: Base Flow Index) against the same indices computed using precipitation and (right) partial correlations of hydrological indices with land cover characteristics (fp: forest probability, fs: forest surface, shs: shrubs surface, ps: pasture surface). Values are expressed in percentage. Significance levels: '·' ≤ 0,1; '*' ≤ 0,05; '**' ≤ 0,01.**

| Hydrological index | Precipitation indices (regression model) | | | Land cover characteristics (partial correlation) | | | |
|---|---|---|---|---|---|---|---|
| | P-M3DMF | P-FRE9 | P-BFI | fp | fs | shs | ps |
| M3DMF | 07 | - | - | * 47 | 10 | ** 69 | 03 |
| FRE9 | - | 05 | - | · 36 | 01 | ** 65 | 04 |
| BFI | - | - | * 53 | 24 | 10 | 10 | 11 |





**Figure 1. Catchments with hydrological records in the study area of the Cantabrian Mountains. The darkest catchments were discarded due to the poor quality of their data.**





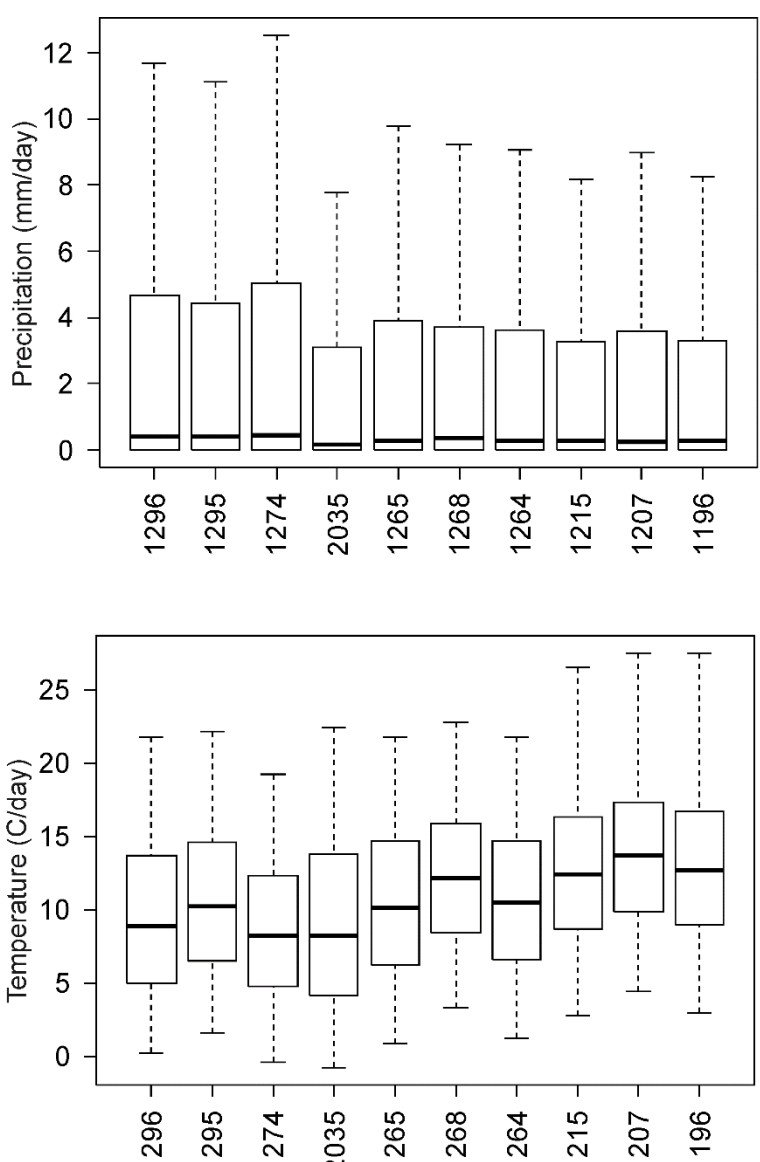

**Figure 2. Daily precipitation and temperature variability for the period 1995-2010 in the 10 catchments of the Cantabrian Mountains, ordered from West (left) to East (right). Boxplots show quartiles. Whiskers show maxima and minima (outliers excluded).**





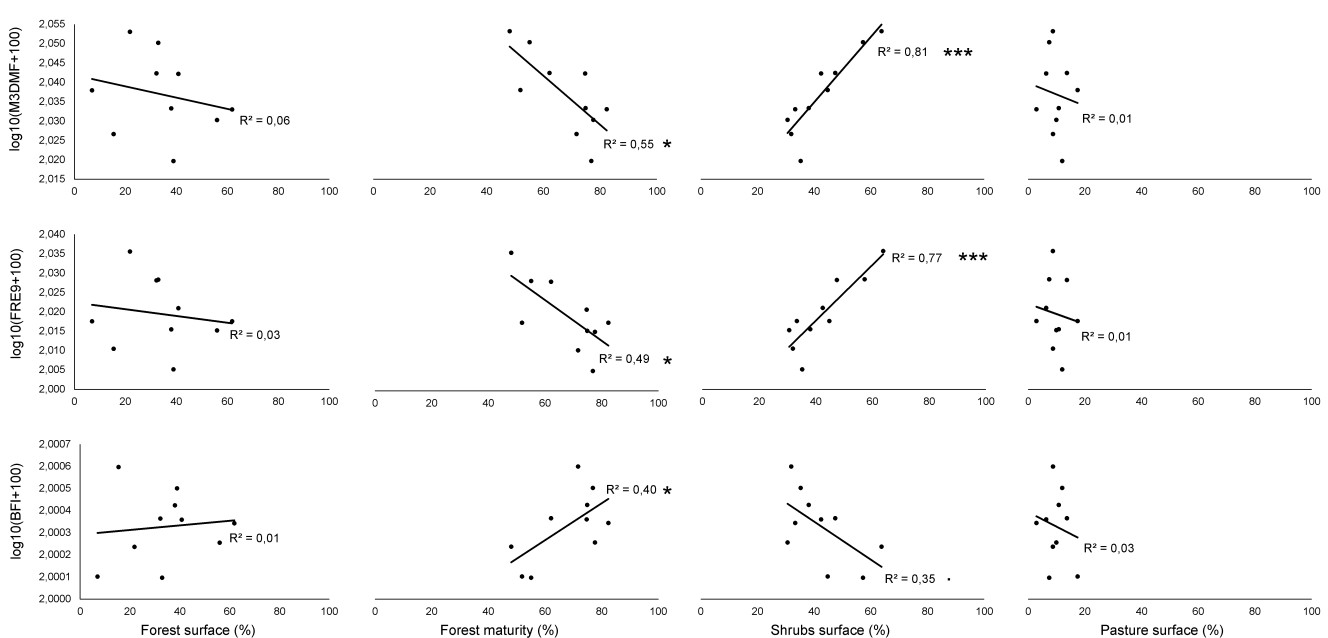

**Figure 3. Regression modelling between land use characteristics and hydrological indices for the period 1995-2010 in the 10 catchments of the Cantabrian Mountains. M3DMF: mean 3-day maximum annual flow, FRE9: number of high flow events per year using an upper threshold of 9 times the median flow over all years, BFI: Base Flow Index. Significance levels: '·' ≤ 0,1; '*' ≤ 0,05; '**' ≤ 0,01; '***' ≤ 0,001.**





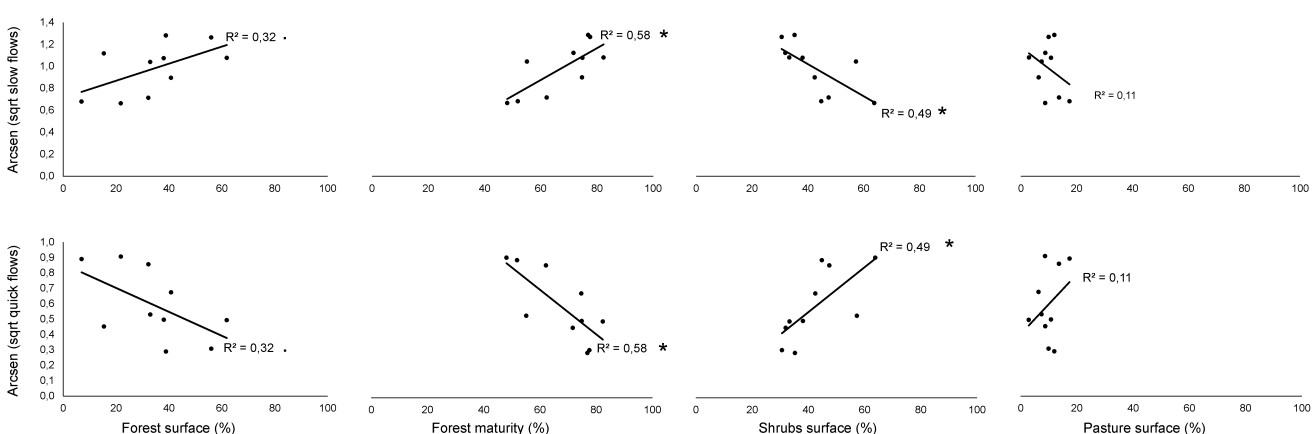

**Figure 4. Regression modelling between land use characteristics and the proportion of slow and quick flows modelled through IHACRES for the period 2000-2007 in the ten catchments of the Cantabrian Mountains. Significance levels: '·' ≤ 0,1; '*' ≤ 0,05.**