# Peer review of "The role of forest maturity on catchment hydrologic stability"

_Hydrology and Earth System Sciences, 2016_

## Referee Comment (RC1) · Anonymous Referee #1 · 3 Nov 2016

General comments: I reviewed the paper "The role of forest maturity on catchment hydrologic stability" by Oscar Belmar and co-workers. In this paper, the authors attempted to improve the understanding of the impact of land cover on flow extremes (flood and drought) at the catchment scale through an experimental design. The objectives and relevant scientific questions addressed in this paper are within the scope of HESS. However, I have to say that the experimental design of using associated with correlations and regression, and speculating that forest maturity can serve as a better hydrological indicator is a little weak. In addition, the authors stated that the first objective of this paper is to isolate the relative contribution of precipitation and land cover to hydrological extreme events. There is model development. If there was an analytical model developed, this would be an adequate contribution. However, this has not been performed in this manuscript. 1. Comment: In the revised version of the

paper, the authors should clearly state which is the novelty of the paper for which the paper deserves publication. 2. Comment: It seems that the authors try to describe some original interpretations for the phenomenon. I would recommend the authors to show the possible mechanisms a little more specifically. That would help particularly the abstract to be more understandable and attractive. 3. Comment: P1, L26, what do you mean by average conditions? 4. Comment: P1, L29, I suggest the authors add the specific forest area. 5. Comment: P2, L20 and L31, what is the difference between watershed and catchment? If this two terms have the same meaning, please used one of them consistently. 6. Comment: In the Introduction section, I could not find detailed research progress of the effect of forest or other land cover on hydrological processes. 7. Comment: The Result section is too short. This again illustrates that the evidence in support of your conclusion is weak and I suggest authors provide more evidence. 8. Comment: The comparison and discussion with the similar studies on the impact of land cover on flow extremes is lack in the manuscript. 9. Comment: P4, L31, how water interception and retention were estimated to determine the proportions of slow and quick flows? Meanwhile, the authors should define what are slow flows and quick flows, respectively. 10. Comment: P5, L9, here the authors only take precipitation into consideration as a climatic factor, how about the effect of evapotranspiration? 11. Comment: P5, L22, what does western extreme mean? 12. Comment: P7, L20-21, I have no idea what you mean here; please improve. 13. Comment: P7, L26, what does the low representation mean? 14. Comment: In Table 1, the authors should provide the mean annual cumulative precipitation and mean annual air temperature, and add what period for hydrological variables (i.e., mean runoff and mean flow), climate variables (i.e., precipitation and temperature) and land cover, though this information have been present in the text body. Furthermore, I suggest that the codes and names of river in eastern and western part of the study area should be distinguished. I wonder that what forest surface is. It refers to vegetation coverage, or something else. What is the relationship between forest surface and forest maturity? 15. Comment: In table 2, partial correlation analysis have been performed between hydrological index (i.e.,

M3DMF, FRE9, and BFI) and forest probability. Yet, in Fig. 3 and Fig. 4, the similar analysis were conducted between hydrological index and forest maturity. Can you please explain this?

———————————————————

---

## Referee Comment (RC2) · Anonymous Referee #2 · 7 Nov 2016

General comments

The present manuscript intends to clarify the influence of land-cover on hydrological regime, particularly in extreme events (floods and droughts). The subject fall within the general scope of "Hydrology and Earth System Sciences" journal. The authors suggest the use of Ordinary Least Square (OLS) regression modelling to related explanatory precipitation and land-use variables to three dependent hydrological variables (3 flow indices). My main concern with the manuscript is that the proposed methodology offers several weaknesses, especially regarding the hydrological modelling and the causal relationships between selected explanatory and depending variables, sometimes circular relations. This makes the conclusions, especially the capability to predict extreme hydrological events based on land-cover characteristics, highly questionable. Also, the Introduction does not provide an appropriate "stat-of-the-art". Authors should deepen

the literature review and clearly describe the objectives of the paper, supported by current knowledge about main drivers of change for extreme hydrological events in temperate Atlantic region.

Specific comments

Introduction: Line 28: ".."land use changes may also affect flood and drought phenology (Scott and Lesch 1997)". What is flood and drought phenology? Also, Scott and Lesch, 1997 study does not address land-cover effects on extreme hydrological events.

Material and Methods 2.2 Land cover characteristics Why did the authors choose to obtain the land-cover information from a supervised classification of a Landsat image and didn't use Corine Land Cover data? The CLC2012 is a free inventory of land cover in 44 classes available for Europe. Forest classes in CLC usually represent mature stages of development and are classified according to forest types (broad-leaved woodlands, coniferous forests, mixed forests). Also, It is not justified why such particular land-use classes have been selected. For instance, what about impervious surfaces? Impervious areas have also impacts in runoff patterns. In my opinion this is more a land-use land-cover (LULC) classification system than only a restrict biophysical description of land type.

Authors should clarify this aspect and include a measure of the accuracy assessment for this supervised classification.

2.4 Hydrological analysis The methodology is not well described: Authors should explain what is considered a "flood" and a "drought" in this study. 10-year flood? 50-year flood? What is an extreme hydrological event in this climate region? And why the three selected flow indices are the most appropriate to characterize "flood" and "droughts" in this climatic region? What are "quick flows" and "slow flows"? Authors should explain the cause-effect with the three flow indices?

Results Authors need to detail the results. What is the percentage of the hydrological

variance that is explained by the precipitation component and by the land-use component? Please give more information in the results section to support our arguments. Probably explore the joint impact of climate and land-cover effects on extreme hydrological events. The poor correlations with the forest surface can be related with some misclassifications errors of the supervised classification?

Discussion I would urge the author to not over-conclude the results of this study regarding those 10 small-catchments.

---

## Author Comment (AC1) · 21 Dec 2016

Dear Prof. Ursino and reviewers,

Thank you very much for the attention received. Please find attached our answer. Should you need further details please do not hesitate to let us know. Yours sincerely,

Oscar Belmar (on behalf of all co-authors).

Please also note the supplement to this comment: http://www.hydrol-earth-syst-sci-discuss.net/hess-2016-471/hess-2016-471-AC1-supplement.zip